# Evaluation of underidentification of potential organ donors in German hospitals

**Grit Esser[1], Benedikt Kolbrink[1], Christoph Borzikowsky[2], Ulrich Kunzendorf[1], Thorsten Feldkamp[1], Kevin Schulte[1] ***

**1** Department of Nephrology and Hypertension, University Hospital Schleswig-Holstein, Christian-Albrechts-University, Kiel, Germany, **2** Institute of Medical Informatics and Statistics, Christian-Albrechts-University, Kiel, Germany

☯ These authors contributed equally to this work.
* Kevin.Schulte@uksh.de

**Data Availability Statement:** Data cannot be shared publicly because of data protection law. The billing data set of German hospitals on which this study is based are available from the Federal

## Abstract

### Background

Since 2010, the number of organ donations in Germany has decreased by one third, mostly due to undetected organ donors. It is unclear, how the undetected potential donor pool is distributed among the different German hospital categories (A = university hospital, B = hospitals with neurosurgery, C = hospitals without neurosurgery) and region types.

### Methods

We performed a nationwide secondary data analysis of all German inpatient cases of the year 2016 (n = 20,063,689). All fatalities were regarded as potential organ donors, in which primary or secondary brain damage was encoded and organ donation was not excluded by a contraindication or a lack of ventilation therapy.

### Results

In 2016, 28,087 potential organ donors were identified. Thereof 21% were found in category A, 28% in category B and 42% in category C hospitals. The contact rate (= organ donation related contacts/ potential organ donors) and realization rate (= realized organ donations/ potential organ donors) of category A, B and C hospitals was 10.6% and 4.6%, 10.9% and 4.8% and 6.0% and 1.7%, respectively. 58.2% of the donor potential of category C hospitals was found in the largest quartile of category C hospitals. 51% (n = 14,436) of the potential organ donors were treated in hospitals in agglomeration areas, 28% (n = 7,909) in urban areas and 21% (n = 5,742) in rural areas. The contact- and realization rate did not significantly differ between these areas.

### Conclusions

The largest proportion of potential organ donors and the lowest realization rate are found in category C hospitals. Reporting and donation practice do not differ between urban and rural regions.

Statistical Office of the Federal Republic of Germany (contact via www.forschungsdatenzentrum.de) for researchers who meet the criteria for access to confidential data.

**Funding:** This study was funded by the Christian-Albrechts-University Kiel, Germany. This funding was awarded to Dr. Kevin Schulte. The funders had no role in study design, data collection and analysis, decision to publish, or preparation of the manuscript.

**Competing interests:** The authors have declared that no competin interests exist.

## Introduction

Since 2010, the number of organ donations in Germany has decreased by one third. This development reached its peak in 2017 when 797 organ donors faced 9,697 patients waiting for a life-saving organ transplant [1]. Accordingly, an increase in organ donation numbers would significantly improve the quality of life [2] and life expectancy of thousands of people [1]. This described decline is mostly due to a reduction of deceased organ donations. While the number of deceased kidney donations dropped in this time from 2,272 to 1,364 by 40%, the number of living kidney donations decreased only slightly from 665 to 557 [1].

Possible reasons for the continuing decrease in organ donations have been discussed extensively in Germany over the last decade [3]. In 2012 it became known that some doctors had falsified their patients' data in a few cases in order to raise their rank on the waiting list. This organ allocation scandal is regarded by many as the main reason for the declining organ donation numbers. They argue that this scandal has a sustainable negative impact on the public's attitude towards organ donation. However, this explanatory approach does not fit in with the fact that the German Federal Centre for Health Education reports a stable and high consent to organ donation in the German population [4]. The number of Germans holding an organ donor card has almost doubled in the last decade [5], which even indicates a rising awareness regarding organ donation, although only a negligible number of the cards are being carried in decisive situations by the patients [6]. In the hope of encouraging more people to make a decision regarding organ donation and thus increase organ donation numbers, the German transplantation law was amended in 2012. Unfortunately, this law neither improved the hospitals' number of contacts to the German Organ Transplantation Foundation (Deutsche Stiftung Organtransplantation, DSO), who is responsible for coordinating organ donation and removal in Germany, nor the number of transplantations in total [7]. The uncertainty among experts and the failure of political interventions underline the need for reliable data to initiate targeted approaches to improve organ donation.

As we reported previously, the number of potential organ donors in Germany has increased steadily since 2010 [8]. This finding ruled out, that the falling donor numbers are an inevitable consequence of an improved treatment of patients with a severe brain damage. We were also able to show that the decrease in organ donations is due to a reporting and recognition deficit of potential organ donors in German hospitals, who bear the core responsibility for reporting potential organ donors to the DSO. Although the number of potential organ donors increased by 13.9% from 2010 to 2015, 18.6% fewer potential organ donors were reported by hospitals [8]. However, important data for the introduction of targeted improvements is still missing.

The German hospitals are divided into three categories by the DSO: University hospitals (category A hospitals), hospitals with an intensive care unit and a department of neurosurgery (category B hospitals), and hospitals with an intensive care unit but without a department of neurosurgery (category C hospitals). To date, it is unclear how the organ donor potential is distributed between these hospital categories and between areas of differing population density. In addition, it is unknown whether the therapeutic procedure differs significantly between hospital categories. Should, for example, invasive ventilation therapy be initiated less frequently in one hospital category, this could have an impact on the organ donor potential of the respective hospitals.

The aim of this study is to create a data basis on which targeted measures can be developed to improve the organ donation process in Germany. Therefore, we analyzed the accounting data of all German inpatient treatment cases of the year 2016 to show the following:

1. The distribution of potential and realized organ donors between the different hospital categories.

2. The proportion of patients who were not eligible as organ donors because no invasive ventilation therapy was initiated depending on the hospital category.

3. The distribution of potential and realized organ donors among urban and rural regions in Germany.

## Materials and methods

### Patients

The German Federal Statistical Office enabled us to analyze the hospital billing data of all German hospitals for the year 2016. This dataset comprises basic patient data, reasons for admission and discharge, main and subsidiary ICD-diagnoses, operation and procedure key codes and information about the duration of an invasive ventilation therapy. To identify potential organ donors, the following four-step selection process was applied (see also Fig 1):

Step 1: Selection of all inpatient cases with death as reason for discharge.

Step 2: Selection of all patients diagnosed with primary or secondary brain damage (inclusion criteria).

Step 3: Exclusion of patients with contraindications for organ donation (exclusion criteria).

Step 4: Exclusion of patients who did not receive invasive ventilation during hospitalization.

This algorithm is based on the "DSO Transplantcheck for Excel" software program [9], which was designed to enable hospitals to analyze retrospectively in which case of death an organ donation would have been, in all probability, possible. Here, these cases are regarded as "potential organ donors". This algorithm was extensively validated in several studies [8,10,11] and showed to be very sensitive but to lack specificity. With regard to the latter, due to the DSO-In-House-Coordination Project [10], in which the suitability of every detected potential organ donor (n = 13,047) to become an actual organ donor was thoroughly evaluated in a case-by-case analysis, it is known that at least 31.7% of the potential organ donors should be reported to the DSO and 10.2% could actually become organ donors.

### Methods

Firstly, hospitals were grouped based on their institution code, so that separate analyses of the different hospital categories were possible. The hospitals were assigned to the different hospital categories based on information published by the DSO [12]. The contact rate (= organ donation related contacts/ potential organ donors) and realization rate (= realized organ donations/ potential organ donors) was calculated for each category and region type. The actual numbers of contacts to the DSO and realized organ donors were taken from the annual report of the DSO for 2016 [13]. Category C hospitals were further sub-classified based on their total number of annual inpatient cases, resulting in the following quartiles: very small hospitals with $\leq$5,395 cases (first quartile), small hospitals with 5,396 to 9,377 cases (second quartile), larger hospitals 9,378 to 14,666 cases (third quartile) and large hospitals with 14,667 to 49,516 cases (fourth quartile). These data were kindly provided by the scientific institute of the statutory health insurance company AOK.

Secondly, the proportion of patients, who were not eligible as organ donors because no invasive ventilation therapy was initiated, was analyzed for the different hospital categories. Therefore, we calculated the rate of potential organ donors and those patients, who died with a main or subsidiary diagnosis of a primary or secondary brain damage and had no contraindications for an organ donation (step 1 to step 3 in the 4-step selection process).

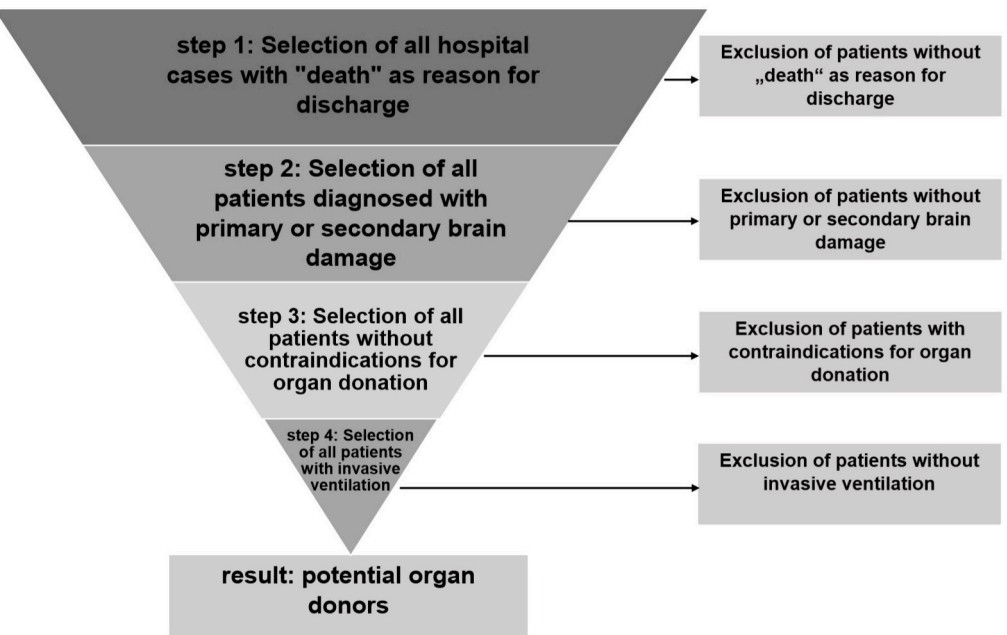

**Fig 1. Analysis algorithm for identification of potential organ donors based on DSO-Transplantcheck for Excel.**

Lastly, we analysed the distribution of the organ donation potential between rural, urban and agglomeration areas. The contact and realization rates were calculated as described above. We defined the different region types in accordance with the German federal institute for construction-, urban- and spatial research [14]. The first region type, rural areas, is defined by a population density of <150 residents/km$^2$ without a regional metropolis with more than 100,000 residents or a population density with <100 residents/km$^2$ with a regional metropolis larger than 100,000 residents. The second region type, urbanized areas, is characterized by a population density of >150 residents/km$^2$ or a regional metropolis with more than 100,000 residents with a minimum population density of ≥100 residents/km$^2$. The third region type, agglomeration areas, is defined by a regional metropolis >300,000 residents or a population density of about 300 residents/km$^2$ (e.g. Berlin, Hamburg or Munich).

IBM SPSS Statistics for Windows (version 22.0.0.2, IBM, 2013) was used to create syntax files and to analyze empirical data provided by The German Federal Statistical Office. Descriptive results will be presented with absolute and relative frequencies. $\chi^2$-test was used to compare categorical variables between different hospital categories and region types. P values <0.05 were regarded as statistically significant.

Our ethics committee waived an evaluation of the study protocol because the routine data was evaluated in anonymized form by the Federal Statistical Office. Since routine patient treatment in Germany is carried out in accordance with the Declaration of Helsinki as well as Istanbul, the present study meets all ethical requirements.

## Results

### Analysis of the organ donor potential depending on the hospital category

In 2016, 20,063,689 inpatients cases were treated in German hospitals in total. Of those cases 416,411 (2.1%) ended with the death of the patient, 16.4% (n = 68,445) of these patients suffered from a primary or secondary brain damage. No absolute contraindication for an organ

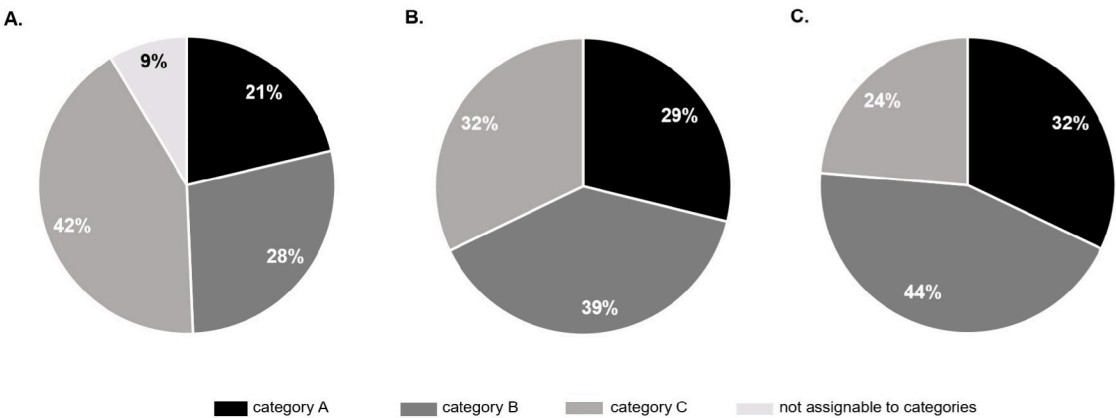

**Fig 2. Distribution of potential organ donors (I.), DSO-contacts (II.) and realized organ donors (III.) among different hospital categories in 2016.** Category A: university hospitals; category B: hospitals with an ICU and a department of neurosurgery; category C: hospitals without a department of neurosurgery.

donation was encoded for 88.6% (n = 60,658) of these patients. 46.3% (n = 28,087) of the remaining cases received an invasive ventilation therapy prior to death and can therewith be regarded as potential organ donors. 21.3% (n = 5,980) of the potential organ donors occurred in category A hospitals, 28.0% (n = 7,878) in category B hospitals, and 42.1% (n = 11,821) in category C hospitals (Fig 2A). 8.6% (n = 2,408) of the potential donors could not be allocated to one specific category, because the accounting data of several hospitals was summarized in one institution number and could not be analyzed separately. We subsumed these cases in the additional category "not assignable to categories" (see Fig 2A). 6.1% (n = 721) of the potential organ donors found in category C hospitals were detected in the first quartile (very small hospitals), 11.3% (n = 1,333) in the second and 24,4% (n = 2,886) in the third quartile. In the fourth quartile (large hospitals) 58.2% (n = 6,881) of the potential organ donors were detected. In this quartile 51.3% (n = 6,302,121) of all cases of category C hospitals were treated (Fig 3).

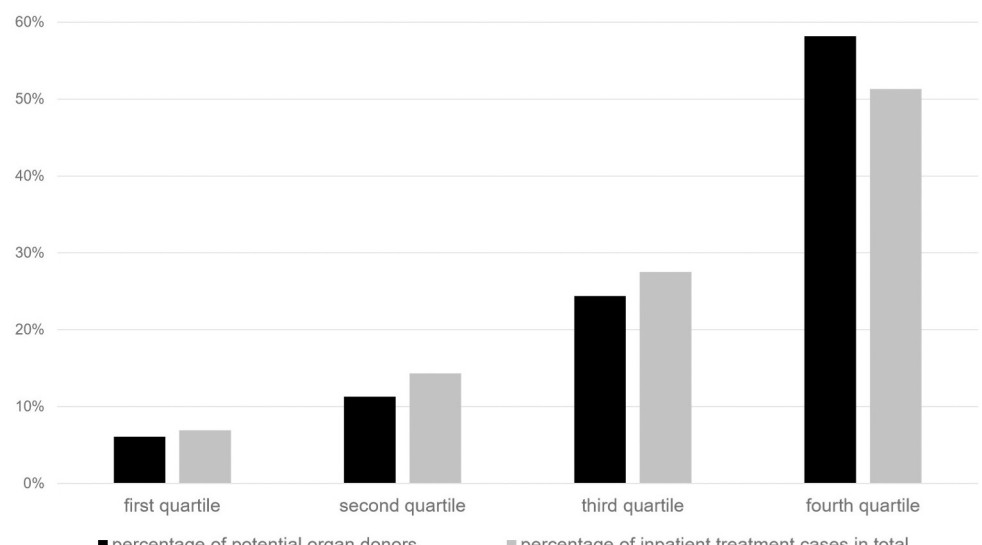

**Fig 3. Distribution of potential organ donors in category C hospitals by hospital size.** The category C hospitals (no department of neurosurgery, n = 932) have been divided into quartiles by their total number of annual inpatient treatment cases (n = 233 per quartile). The first quartile includes hospitals ≤5,395 cases, the second quartile hospitals with 5,396 to 9,377 cases, the third quartile hospitals with 9,378 to 14,666 cases and the fourth quartile hospitals with 14,667 up to 49,516 cases.

In 2016, the DSO registered 2,193 contacts. Of these contacts, 28.8% (n = 632) were initiated by category A hospitals, 39.0% (n = 856) by category B hospitals and 32.1% (n = 705) by category C hospitals (Fig 2B). This results in a contact rate of 10.6% in category A, 10.9% in category B, 6.0% in category C hospitals and a mean contact rate of 7.8% over all categories. Of the 857 organ donors realized in the reference year, 32.1% (n = 275) were realized in category A, 44.2% (n = 379) in category B and 23.7% (n = 203) in category C hospitals (Fig 2C). The mean realization rate was 3.1%, with 4.6% in category A, 4.8% in category B and 1.7% in category C hospitals. The contact- and realization rates of category C hospitals were significantly lower than those of category A or B hospitals (p<0.001).

## Comparison of the proportion of invasively ventilated potential organ donors between the different hospital categories

In category A hospitals we found 8,595 cases of death with a main or subsidiary diagnosis of primary or secondary brain damage without a contraindication for organ donation. Of these, 30.4% (n = 2,615) did not receive invasive ventilation and, thus, could not be classified as potential organ donors. In category B hospitals 14,017 patient cases had an encoded brain damage and no contraindication for organ donation, of those 43.8% (n = 6,139) received no invasive ventilation. Lastly, in category C hospitals 32,111 fatalities with an encoded primary or secondary brain damage and without contraindications were detected. Of those, 63.2% (n = 20,290) did not receive an invasive ventilation therapy (Fig 4).

## Analysis of the organ donation potential with regard to region type

In the agglomeration area 51.4% (n = 14,436) of the potential organ donors were found, 51.3% (n = 1,124) of all contacts initiated and 51.0% (n = 437) of all organ donations realized. In the urbanized area we found 28.2% (n = 7,909) of the potential organ donors, 26.9% (n = 589) of all contacts with the DSO and 26.4% (n = 226) of the realized organ donations. In the rural area 20.4% (n = 5,742) of the potential organ donors were located, 21.8% (n = 480) of all DSO contacts happened and 22.6% (n = 194) of all organ donations were performed (Fig 5A–5C). The contact- and realization rates were 7.8% and 3.0% for the agglomeration area, 7.4% and 2.9% for the urbanized area, 8.4% and 3.4% in the rural area and did not differ significantly between the region types.

## Discussion

The present study has three major findings:

1. The highest proportion of potential organ donors can be found in category C hospitals. These hospitals do have a significant lower contact- and realization rate than category A and B hospitals.

2. The proportion of patients with severe brain damage receiving invasive ventilation prior to their death is considerably larger in category A and B hospitals than in the category C hospitals.

3. Contact- and realization rates do not differ significantly among urban and rural regions in Germany.

Until today, the thesis has been frequently put forward that potential organ donors are mainly to be found in large hospitals [15]. Following this assertion, it seems reasonable to concentrate efforts to increase organ donation numbers mainly in those large centers. Our study confirms that the number of potential organ donors per hospital is indeed highest in category

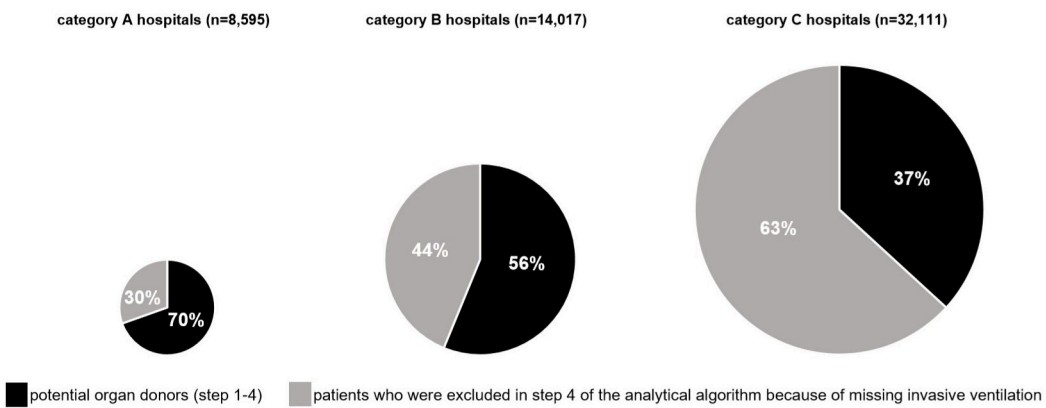

**Fig 4. Comparison of the proportion of invasively ventilated patients with severe brain damage between the different hospital categories.**

A and B hospitals. However, the largest proportion of the national organ donation potential and especially the most unidentified potential organ donors are found in category C hospitals. This finding is supported by a smaller retrospective analysis recently performed in the eastern region of Germany, which showed that 69.8% of all potential organ donors can be found in category C hospitals [16]. This is an important finding making clear that a significant increase in organ donation in Germany requires strategies that above all address the identification and reporting deficit of potential organ donors in category C hospitals. Since the majority of potential organ donors can be found in the largest quartile of category C hospitals, it seems justifiable and probable more effective, to focus on those larger category C hospitals. If the realization rate in these hospitals (2016: 1.7%) could be increased only to the level currently achieved in category A hospitals (2016: 4.6%), the number of organ donations in Germany could be increased by more than one third. Nevertheless, the especially low contact- and realization rates in category C hospitals should not detract from the fact that there is much room for improvement in all categories. If all hospitals achieved a contact rate of 30% and a realization rate of 10% of the potential organ donors, which seems to be realistically achievable based on the results of the DSO-Inhouse coordination project [10], the number of organ donors could be increased to about 2,800 per year.

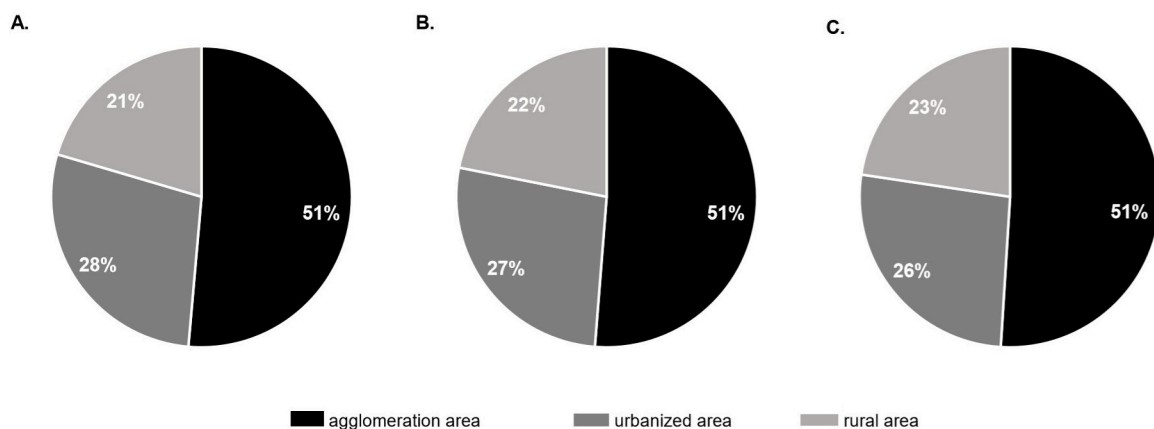

**Fig 5. Distribution of potential organ donors (I.), DSO-contacts (II.) and realized organ donors (III.) among different region types in 2016.**

What has to be done now? In a preliminary study we were able to show that the contact- and realization rates of different hospitals vary markedly. Although the number of potential organ donors did not differ between these hospitals, the number of realized organ donors differed in some cases by more than a factor of 15 [8]. It is important to realize that we found these big differences between hospitals, which all had a full-time transplant coordinator and received the same compensation for a realized transplant. Because some hospitals were able to realize a very high donation rate, the causes cannot be of a general nature but must exist at the hospital level. Unfortunately, up to now it was not possible to analyze the individual strength and weaknesses of these hospitals. Therefore, no scientifically based general answer can be provided to the question how the untapped potential can be utilized most effectively. Due to this it is crucial to identify individual weaknesses and problems of the hospitals to find starting points for effective improvements. This process is an important part of the quality surveillance- and improvement system in countries with much higher organ donation rates per million population than Germany, for example Spain [17] or the USA [18]. Up to now, such procedures were not feasible in Germany, since they require disclosure and analysis of the transplantation related data of the respective hospitals and this data has not been publicly available so far. Last year, however, there were significant changes in this regard: In April 2019, a further revision of the German transplantation law stipulated, in addition to a number of other points, that every hospital must now report its organ donation potential annually and give an account of why potential organ donors were not reported. These data will be available to the DSO for the first time in autumn 2020 and then be published [19]. So far, no detailed rules have been laid down on how to proceed if a hospital reports only a very small percentage of potential organ donors. Our study shows that the public and regulatory authorities should focus on the large category C hospitals in particular to guarantee that a detailed root-cause analysis is initiated in these hospitals.

The second main finding of our study is that the proportion of patients with severe brain damage who receive invasive ventilation differs significantly between different hospital categories. While 70% of the patients in university hospitals received invasive ventilation before their death, only 37% did in category C hospitals. Even if other patients are treated in specialized university hospitals than in category C hospitals, we believe that this finding cannot be attributed solely to this. The reasoning for our assumption is that more than 707 ICD codes were used to select the corresponding patients. This makes it likely that patient characteristics in the different hospital categories are at least similar. Furthermore, a study by Brauer and colleagues using the same algorithm in 144 hospitals in Germany suggests that there are also no major differences in terms of age of the patients [16]. Based on this, it seems likely that the treatment of patients with neurological diseases is indeed more often terminated early in category C hospitals. There are several possible explanations for this state of affairs. It could be due to the fact that they have less intensive care capacity than the other hospital categories and are therefore more reluctant to initiate ventilation therapy if the prognosis for the patient is poor. Another explanation could be that they lack competence in treating these cases because they do not have a neurosurgical department. However, this probably has a considerable influence on the quality of care for these patients and urgently calls for further research. Although we can only speculate why therapy in category C hospitals is terminated earlier than in the hospitals categories, our result at least allows the assumption that the organ donor potential in category C hospitals could be even greater as our above-mentioned analysis suggests.

Addressing the last of our main findings, there was no difference in contact- and realization rates between rural and urban regions. Investigations of this kind have—at least to our knowledge—never been conducted in Germany before. Significant differences between rural- and urban organ donors and recipients in the United States have been demonstrated in the past

[20,21]. Thus, we generated the thesis that there could be a disparity between rural, more remote hospitals and the ones in urban areas. By this work we disprove our assumption and show that—in Germany at least—the location of a hospital does not affect its performance regarding the recruitment of potential organ donors. Accordingly, there is no need for measures aimed specifically at rural areas.

In summary, our study reveals that it would be most useful to develop measures to improve the identification and reporting of potential organ donors in large category C hospitals, because there is a particularly large, undetected donor potential. Since all measures to increase the organ donation rate presuppose an unused organ donor potential, we believe that this result is of central importance in order to effectively and purposefully counteract the falling organ donation numbers in Germany. As our study describes a methodical approach to identify the unused national organ donation potential, we believe that it is also of great international interest.

## Strengths and limitations

The strength of our study is the fact that all hospital inpatient cases of the year 2016 were integrated into the analysis. By this approach the study creates an overall picture of the situation in Germany and does not run the risk of being distorted by a selection bias.

Nevertheless, a major limitation of our study is its retrospective approach and the fact that it is based on primary data that were not originally collected to answer our questions. As we have used an extensively validated algorithm, we nevertheless consider the number of identified potential organ donors to be reasonable. When speaking of "potential organ donors" internationally, it should be taken into consideration that these are often defined as patients, who were found suitable for organ donation based on clinical findings. Therefore, the numbers of potential donors identified in our study are likely to be higher than those in similar international studies [22,23].

Lastly, our study is limited by the fact that the primary data set depends on the coding quality of the hospitals. In the German DRG-system a correct coding of secondary diagnoses leads to an increase in the case value of a patient treatment. For this reason, there is a financial incentive for hospitals to code relevant secondary diagnoses (such as tumor diseases). In return, there is an incentive for the Medical Service of the health insurance companies to prevent over-coding for the same reason. Therefore, we assume that the quality of the data set on which the study is based is very solid as it is under control from two sides.

## Author Contributions

**Conceptualization:** Grit Esser, Benedikt Kolbrink, Thorsten Feldkamp, Kevin Schulte.

**Data curation:** Grit Esser, Benedikt Kolbrink, Kevin Schulte.

**Formal analysis:** Grit Esser, Benedikt Kolbrink, Christoph Borzikowsky, Thorsten Feldkamp, Kevin Schulte.

**Funding acquisition:** Kevin Schulte.

**Investigation:** Grit Esser, Benedikt Kolbrink, Kevin Schulte.

**Methodology:** Kevin Schulte.

**Project administration:** Ulrich Kunzendorf, Thorsten Feldkamp, Kevin Schulte.

**Resources:** Kevin Schulte.

**Software:** Kevin Schulte.

**Supervision:** Christoph Borzikowsky, Ulrich Kunzendorf, Thorsten Feldkamp, Kevin Schulte.

**Validation:** Kevin Schulte.

**Visualization:** Grit Esser, Benedikt Kolbrink, Kevin Schulte.

**Writing – original draft:** Grit Esser, Benedikt Kolbrink, Kevin Schulte.

**Writing – review & editing:** Christoph Borzikowsky, Ulrich Kunzendorf, Thorsten Feldkamp, Kevin Schulte.

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
