## [Decision Letter · Decision Letter 0]

22 Jul 2020

PONE-D-20-20570

Evaluation of the undetected organ donor potential in Germany

PLOS ONE

Dear Dr. Schulte,

Thank you for submitting your manuscript to PLOS ONE. After careful consideration, we feel that it has merit but does not fully meet PLOS ONE’s publication criteria as it currently stands. Therefore, we invite you to submit a revised version of the manuscript that addresses the points raised during the review process.

We look forward to receiving your revised manuscript.

Kind regards,

Larry Allan Weinrauch, MD

Academic Editor

PLOS ONE

Journal Requirements:

2.We note that you have indicated that data from this study are available upon request. PLOS only allows data to be available upon request if there are legal or ethical restrictions on sharing data publicly. For information on unacceptable data access restrictions, please see http://journals.plos.org/plosone/s/data-availability#loc-unacceptable-data-access-restrictions.

Additional Editor Comments (if provided):

The current manuscript requires revision in line with the comments of the reviewers. Proper revision will improve the readability of this work which demonstrates a difference between harvesting of appropriate organs by type of institution. The paper would be more impactful if it concluded with possible solutions that might improve organ harvest (donation)

Reviewers' comments:

Reviewer's Responses to Questions

**Comments to the Author**

1. Is the manuscript technically sound, and do the data support the conclusions?

Reviewer #1: Yes

Reviewer #2: Yes

2. Has the statistical analysis been performed appropriately and rigorously? 

Reviewer #1: Yes

Reviewer #2: Yes

3. Have the authors made all data underlying the findings in their manuscript fully available?

Reviewer #1: Yes

Reviewer #2: Yes

4. Is the manuscript presented in an intelligible fashion and written in standard English?

Reviewer #1: Yes

Reviewer #2: Yes

5. Review Comments to the Author

Reviewer #1: The fact that the highest proportion of potential organ donors can be found in category C hospitals is of paramount practical importance for addressing the problem of decreasing deceased donors in future.

All over the manuscript the word donor has been implied to cadaver donors in Germany and for an international reader, it would be helpful to know what percentage of the total donors in Germany come from live donation and what is the temporal trends of that. In some regions of the world , increasing number of live donations, that have better outcomes, accounts for some of the decreased deceased donations.

Reviewer #2: In their paper Evaluation of the undetected organ donor potential in Germany, Esser et al examined data from the German Federal Statistical Office to evaluate number of potential organ donors at German hospitals of different sizes and location and compared that to data from the German Foundation for organ donation.They found that the number of unidentified potential organ donors was greatest at so-called category C hospitals - those with ICUs but no neurological service - significantly in the largest quartile of category C hospitals. They found no diffidence in unidentified potential organ donors based on population size of the regions service by the hospitals. They concluded that about a third of the unmet demand for kidney transplants could be better met by better identification of potential donors in the larger category C hospitals. Their paper is interesting but raises several questions that need to be addressed.

1. Whose responsibility is it to identify potential organ donors? Is the problem that insufficient attention is being paid by hospital staff to notifying the DSO. Or is the problem that the DSO lacks the resources to perform outreach to these hospitals. The paper needs a little more background on the usual practices by which potential organ donors are identified and how their donation is realized.

2. The authors argue that the decline in organ donation is due to the number of unidentified potential donors. But they only provide one year of data and don’t show that that this number is growing. While their paper doesn’t make the case that this is the reason, it does seek to show that improving identification of potential donors would be a way to slow the decline.

3. Why were patients with brain injury not on mechanical in greater numbers at category C hospitals. Since the category C hospitals don’t have neurosurgery departments, is the implication was that these patients were not considered candidates for mechanical ventilation because their chances for recovery of meaningful brain function were considered poor? Is there any mechanism in Germany for transferring these patients to an A or B category hospital for neurosurgery evaluation? This is perhaps beyond the scope of the paper but gets to me next question...

4. What are the solutions for tapping this underutilized potential source of organ donors and what efforts are underway to do that. In my limited experience taking organ call at a US hospital, I often receive organ offers from potential donors who presented to small hospital, were identified with severe head trauma, intubated and sent to a larger center for neurosurgery evaluation. They are considered inoperable candidates and are referred to the organ procurement organization as potential organ donors. What is the system in place in Germany?

5. The title of the article needs to reflect the substance of the article. I would suggest something like “Evaluation of underidentification of potential organ donors in German Hospitals.”

6. At several points early in the paper, the authors refer to the different categories of hospitals as “clinics.” To avoid confusion and maintain consistency, I would refer throughout the paper to category A, B and C hospitals.

7. On line 101, the authors refer to the organ scandal of 2012 as a potential reason for the decline, but then without providing any background on the scandal, they say this doesn’t fit with the facts. Maybe better to take this reference out if it isn’t relevant. Otherwise they need some explanation of the scandal.

8. Why did they look at only one year’s worth of data, and why not chose a more recent year?

9. Line 191 please confirm the direction of the arrow.Are rural population really defined as “great than” 150 residents per square kilometer? Or should that be less than

10. Please define the term agglomeration for readers who are not demographers, or provide an example of such a region in Germany

6. PLOS authors have the option to publish the peer review history of their article (what does this mean?). If published, this will include your full peer review and any attached files.

Reviewer #1: **Yes: **Bijan Roshan

Reviewer #2: No

---

## [Author Response · Author response to Decision Letter 0]

17 Sep 2020

Additional Editor Comments (if provided):

The current manuscript requires revision in line with the comments of the reviewers. Proper revision will improve the readability of this work which demonstrates a difference between harvesting of appropriate organs by type of institution. The paper would be more impactful if it concluded with possible solutions that might improve organ harvest (donation)

We have taken great efforts to revise and improve our manuscript based on the comments of the reviewers. In particular we have tried to make the manuscript easier to understand for international readers. 

Furthermore, we have tried to work out more clearly what we think must happen now in order to increase the number of organ donations in Germany. Therefore, we have integrated following new passages into the discussion section: 

p.14, lines 321ff 

“What has to be done now? In a preliminary study we were able to show that the contact- and realization rates of different hospitals vary markedly. Although the number of potential organ donors did not differ between these hospitals, the number of realized organ donors differed in some cases by more than a factor of 15 (8). It is important to realize that we found these big differences between hospitals, which all had a full-time transplant coordinator and received the same compensation for a realized transplant. Because some hospitals were able to realize a very high donation rate, the causes cannot be of a general nature but must exist at the hospital level. Unfortunately, up to now it was not possible to analyze the individual strength and weaknesses of these hospitals. Therefore, no scientifically based general answer can be provided to the question how the untapped potential can be utilized most effectively. Due to this it is crucial to identify individual weaknesses and problems of the hospitals to find starting points for effective improvements. “

p.15, lines 343ff 

“So far, no detailed rules have been laid down on how to proceed if a hospital reports only a very small percentage of potential organ donors. Our study shows that the public and regulatory authorities should focus on the large category C hospitals in particular to guarantee that a detailed root-cause analysis is initiated in these hospitals. “ 

Reviewers' comments:

Reviewer's Comments to the Author

Reviewer #1: The fact that the highest proportion of potential organ donors can be found in category C hospitals is of paramount practical importance for addressing the problem of decreasing deceased donors in future

All over in the manuscript the word donor has been implied to cadaver donors in Germany and for an international reader, it would be helpful to know what percentage of the total donors in Germany come from live donation and what is the temporal trends of that. In some regions of the world, increasing number of live donations, that have better outcomes, accounts for some of the decreased deceased donations.

Since 2010, the number of living kidney donations has decreased in Germany from 665 (22.6% of all kidney donations) to 557 (29% of all kidney donations) in 2017. Therefore, a rising number of living kidney donations cannot be made responsible for the falling donation numbers in Germany. 

We included this important aspect into our manuscript at page 4, line 91ff: 

„This described decline is mostly due to a reduction of deceased organ donations. While the number of deceased kidney donations dropped in this time from 2,272 to 1,364 by 40%, the number of living donations decreased only slightly from 665 to 557 [1].“ 

Reviewer #2: In their paper Evaluation of the undetected organ donor potential in Germany, Esser et al examined data from the German Federal Statistical Office to evaluate number of potential organ donors at German hospitals of different sizes and location and compared that to data from the German Foundation for organ donation.They found that the number of unidentified potential organ donors was greatest at so-called category C hospitals - those with ICUs but no neurological service - significantly in the largest quartile of category C hospitals. They found no difference in unidentified potential organ donors based on population size of the regions service by the hospitals. They concluded that about a third of the unmet demand for kidney transplants could be better met by better identification of potential donors in the larger category C hospitals. Their paper is interesting but raises several questions that need to be addressed.

1. Whose responsibility is it to identify potential organ donors? Is the problem that insufficient attention is being paid by hospital staff to notifying the DSO. Or is the problem that the DSO lacks the resources to perform outreach to these hospitals. The paper needs a little more background on the usual practices by which potential organ donors are identified and how their donation is realized.

In Germany, hospitals are responsible for identifying and reporting potential organ donors to the DSO. The DSO, on the other hand, is responsible for accompanying the interviews with the relatives and organising the explantation. 

We have changed the following text passages in order to make the task of the DSO and the hospitals in Germany clearer: 

p. 4, lines 110-111: 

“…, who is responsible for coordinating organ donation and removal in Germany, …”

p. 5, lines 118ff.: 

„We were also able to show that the decrease in organ donations is due to a reporting and recognition deficit of potential organ donors in German hospitals, who bear the core responsibility for reporting potential organ donors to the DSO.“

2. The authors argue that the decline in organ donation is due to the number of unidentified potential donors. But they only provide one year of data and don’t show that that this number is growing. While their paper doesn’t make the case that this is the reason, it does seek to show that improving identification of potential donors would be a way to slow the decline.

In a much acclaimed preliminary study we were able to show that the number of potential organ donors in Germany increased by 13.9% from 2010 to 2015, while the number of contacts between hospitals and the DSO decreased by 18.6% (Schulte K, et al. Decline in Organ Donation in Germany. Deutsches Arzteblatt international. 2018). 

We have improved the following passage to clarify this aspect: 

p.5, lines 120ff

„Although the number of potential organ donors increased by 13,9% from 2010 to 2015, 18.6% fewer potential organ donors were reported by hospitals [8].“ 

3. Why were patients with brain injury not on mechanical in greater numbers at category C hospitals? Since the category C hospitals don’t have neurosurgery departments, is the implication was that these patients were not considered candidates for mechanical ventilation because their chances for recovery of meaningful brain function were considered poor? Is there any mechanism in Germany for transferring these patients to an A or B category hospital for neurosurgery evaluation? This is perhaps beyond the scope of the paper but gets to me next question...

This is a very interersting and important question. Unfortunately, our analysis algorithm does not allow us to answer this point properly. Of course, there is the possibility in Germany to transfer patients with severe neurological diseases from a category C hospital to a category A or B hospital. However, there are no national guidelines and no traceable documentation for these cases. In our view, this finding points to a serious quality problem in the care of neurological and neurosurgical diseases in category C hospitals. However, further investigations are necessary to characterize this in more detail. 

4. What are the solutions for tapping this underutilized potential source of organ donors and what efforts are underway to do that. In my limited experience taking organ call at a US hospital, I often receive organ offers from potential donors who presented to small hospital, were identified with severe head trauma, intubated and sent to a larger center for neurosurgery evaluation. They are considered inoperable candidates and are referred to the organ procurement organization as potential organ donors. What is the system in place in Germany?

In Germany, every hospital (categories A, B and C) must appoint a transplant coordinator. Since the last amendment to the transplant law in 2019, it has been ensured that the transplant coordinators are at least partially released from their other clinical activities for this purpose. In addition, a nationwide neurological consultation service was established in 2019 to ensure that brain death diagnosis can be carried out promptly in rural regions as well. As our investigation now shows, this measure would probably not have been necessary, as the identification of potential organ donors in rural regions is not worse than in urban areas. Furthermore, since 2019 hospitals are paid considerably better when they make organ donations. Despite all these measures, the number of organ donors has not increased in 2019 compared to 2018 but decreased, from 955 to 932 donors. 

Interestingly, we were able to show in a preliminary study (reference year 2015) that the number of organ donors differs between different German hospitals in some cases by more than a factor of 15. Of note, the number of potential organ donors did not differ between these hospitals! It is important to note that there was a full-time transplant coordinator in each of these hospitals. Furthermore, all hospitals were compensated equally for their work, so this cannot explain the differences either. Unfortunately, due to considerable resistance, we have not yet been able to go beyond an analysis of the accounting data. It is therefore not known why one hospital reports almost all potential organ donors and the other almost none to the DSO. For this reason, we believe that an external quality assurance is necessary, which fortunately was also initiated by German politicians in 2018 (see p.14, lines 339ff). In autumn of this year, it will be publicly visible for the first time, which hospitals have not reported their potential organ donors to the DSO. We hope that the present study will help to ensure that the focus of politicians and decision-makers is then directed to where the greatest untapped potential can be found. 

We have revised the first passage of our discussion to point out this issue more deeply (p.14, lines 322ff, p.15, lines 344ff). 

5. The title of the article needs to reflect the substance of the article. I would suggest something like “Evaluation of underidentification of potential organ donors in German Hospitals”.

Thank you very much for this suggestion, which we have gladly taken over.

6. At several points early in the paper, the authors refer to the different categories of hospitals as “clinics”. To avoid confusion and maintain consistency, I would refer throughout the paper to category A, B and C hospitals.

We have now consistently used the term "A, B or C hospital" in our manuscript.

7. On line 101, the authors refer to the organ scandal of 2012 as a potential reason for the decline, but then without providing any background on the scandal, they say this doesn’t fit with the facts. Maybe better to take this reference out if it isn’t relevant. Otherwise they need some explanation of the scandal.

In 2012, it became known that a few doctors had falsified their patients' records to increase their rank on the waiting list. This malpractice received a great deal of public attention in Germany. Politicians, doctors and association officials have repeteadly seen this scandal as the reason for the declining organ donation figures in Germany. 

Although this explanation may seem plausible at first glance, several findings clearly indicate that this scandal cannot be the leading cause of the declining numbers: 

1.) The organ donation figures had already fallen significantly in the two previous years (2010 and 2011).

2.) No change in the attitude of the population could be measured after the scandal, although the Federal Centre for Health Education regularly conducts major studies on this issue. 

In our opinion, it is very important to make it clear once again that the declining organ donation numbers in Germany are not an expression of a change in the attitude of the population but are due to process problems in German hospitals. Therefore, we have not deleted this section, but have revised it to make the background more understandable. This passage now reads following:

p. 4, lines 96ff

„In 2012 it became known that some doctors had falsified their patients` data in a few cases in order to gain an advantage for them on the waiting list. This organ allocation scandal is regarded by many as the main reason for the declining organ donation numbers. They argue that this scandal has a sustainable negative impact on the public`s attitude towards organ donation.“

8. Why did they look at only one year’s worth of data, and why not chose a more recent year?

In a previous study (Schulte K, et al. Decline in Organ Donation in Germany. Deutsches Arzteblatt international. 2018.) we analysed the accounting data for the years 2010 to 2015 and looked at longitudinal changes. The main goal of this study was to examine the distribution of potential organ donors within one year, which is why we concentrated on the analysis year 2016.

On the second point: The German Federal Statistical Office receives the hospital billing data after a significant delay, which is why the data set for 2017 is still not available for scientific purposes. Therefore, we were not able to analyze the data from 2017 or even 2018. 

9. Line 191 please confirm the direction of the arrow. Are rural population really defined as “great than” 150 residents per square kilometer? Or should that be less than.

Thank you for this hint! We corrected the direction ot the arrow (p.8, line 196). 

10. Please define the term agglomeration for readers who are not demographers, or provide an example of such a region in Germany

We added some examples for such a region in Germany to make the term more comprehensible. 

p.9, line 204:

„The third region type, agglomeration areas, is defined by a regional metropolis >300,000 residents or a population density of about 300 residents/km² (e.g. Berlin, Hamburg or Munich).“

---

## [Decision Letter · Decision Letter 1]

9 Nov 2020

Evaluation of underidentification of potential organ donors in German hospitals

PONE-D-20-20570R1

Dear Dr. ,

We’re pleased to inform you that your manuscript has been judged scientifically suitable for publication and will be formally accepted for publication once it meets all outstanding technical requirements.

Kind regards,

Larry Allan Weinrauch, MD

Academic Editor

PLOS ONE

Additional Editor Comments (optional):

It will be interesting to see their followup once more data becomes available from the DSO and whether efforts yield an increase in donations from the largest category C hospitals.

It would be nice too see a chart with the differences in live related, live unrelated and cadaveric grafts

Reviewers' comments:

Reviewer's Responses to Questions

**Comments to the Author**

1. If the authors have adequately addressed your comments raised in a previous round of review and you feel that this manuscript is now acceptable for publication, you may indicate that here to bypass the “Comments to the Author” section, enter your conflict of interest statement in the “Confidential to Editor” section, and submit your "Accept" recommendation.

Reviewer #1: All comments have been addressed

Reviewer #2: All comments have been addressed

2. Is the manuscript technically sound, and do the data support the conclusions?

Reviewer #1: Yes

Reviewer #2: Yes

3. Has the statistical analysis been performed appropriately and rigorously? 

Reviewer #1: Yes

Reviewer #2: Yes

4. Have the authors made all data underlying the findings in their manuscript fully available?

Reviewer #1: Yes

Reviewer #2: Yes

5. Is the manuscript presented in an intelligible fashion and written in standard English?

Reviewer #1: Yes

Reviewer #2: Yes

6. Review Comments to the Author

Reviewer #1: I am satisfied with the answers. Still, it would be nice to have a figure to show the changes in deceased vs live donations in Germany in the comparison period.

Reviewer #2: The authors have addressed my concerns and comments fully. It will be interesting to see their followup once more data becomes available from the DSO and whether efforts yield an increase in donations from the largest category C hospitals.

7. PLOS authors have the option to publish the peer review history of their article (what does this mean?). If published, this will include your full peer review and any attached files.

Reviewer #1: **Yes: **Bijan Roshan, MD

Reviewer #2: No

---

## [Editor Report · Acceptance letter]

11 Nov 2020

PONE-D-20-20570R1 

Evaluation of underidentification of potential organ donors in German hospitals 

Dear Dr. Schulte:

I'm pleased to inform you that your manuscript has been deemed suitable for publication in PLOS ONE. Congratulations! Your manuscript is now with our production department. 

Kind regards, 

on behalf of

Dr. Larry Allan Weinrauch 

Academic Editor

PLOS ONE